# Comparative Analysis of Epididymis Cauda of Yak before and after Sexual Maturity

**DOI:** 10.3390/ani13081355

**Published:** 2023-04-15

**Authors:** Ziqiang Ding, Lin Xiong, Xingdong Wang, Shaoke Guo, Mengli Cao, Yandong Kang, Yongfu La, Pengjia Bao, Jie Pei, Xian Guo

**Affiliations:** 1Key Laboratory of Yak Breeding Engineering of Gansu Province, Lanzhou Institute of Husbandry and Pharmaceutical Sciences, Chinese Academy of Agricultural Sciences, Lanzhou 730050, China; dingziqiang1997@163.com (Z.D.); xionglin@caas.cn (L.X.); wxd17339929758@163.com (X.W.); gsk1125@163.com (S.G.); caomengliaaa@163.com (M.C.); kangyandong0901@163.com (Y.K.); layongfu@caas.cn (Y.L.); baopengjia@caas.cn (P.B.); 2Key Laboratory of Animal Genetics and Breeding on Tibetan Plateau, Ministry of Agriculture and Rural Affairs, Lanzhou 730050, China

**Keywords:** epididymis, proteomics, RNA-seq, sperm maturation, yak

## Abstract

**Simple Summary:**

Yak are an important source of produce and living materials for plateau herdsmen. However, the population of yak is small and their reproductive capacity is low, so it is very important to improve their reproductive potential. The epididymis is an important place for sperm maturation and storage, but the differential expression of the cauda epididymis before and after sexual maturity in yak has not been characterized. In this study, the key genes regulating epididymal cauda development and sperm maturation were screened by RNA-seq and proteomics. These results ultimately provide a theoretical basis for how to improve the reproductive potential of yak.

**Abstract:**

Epididymis development is the basis of male reproduction and is a crucial site where sperm maturation occurs. In order to further understand the epididymal development of yak and how to regulate sperm maturation, we conducted a multi-omics analysis. We detected 2274 differential genes, 222 differential proteins and 117 co-expression genes in the cauda epididymis of yak before and after sexual maturity by RNA-seq and proteomics techniques, which included TGFBI, COL1A1, COL1A2, COL3A1, COL12A1, SULT2B1, KRT19, and NPC2. These high abundance genes are mainly related to cell growth, differentiation, adhesion and sperm maturation, and are mainly enriched via extracellular matrix receptor interaction, protein differentiation and absorption, and lysosome and estrogen signaling pathways. The abnormal expression of these genes may lead to the retardation of epididymal cauda development and abnormal sperm function in yak. In conclusion, through single and combined analysis, we provided a theoretical basis for the development of the yak epididymal cauda, sperm maturation, and screening of key genes involved in the regulation of male yak reproduction.

## 1. Introduction

The epididymis is composed of a series of tubular structures and is a crucial site for sperm development, modification, maturation and storage. Anatomy has revealed that the epididymis can be divided into three parts: caput, corpus, and cauda [1,2]. Specific transcription factors and proteins present in different parts of the epididymis create different lumen microenvironments. When the sperm passes through, it can modify the lipid composition of the sperm plasma membrane and regulate the surface protein composition, so as to complete continuous sperm modification and make the sperm obtain motility and fertilization [3,4,5]. A crucial place for sperm storage, the cauda of the epididymis can protect sperm from oxidative damage, maintain its fertility, and remain still [6,7]. Therefore, exploration of this transcript, along with proteins in the epididymal cauda, is vital for understanding the mechanism underlying sperm maturation.

The emergence of sequencing technology marks the transition of biology from cell level to molecular level and greatly promotes research into the molecular mechanism. At present, the epididymis of mice, rats, cattle and humans has been analyzed and described by RNA-seq technology [8,9,10,11]. Deep mRNA sequencing of human epididymal cells has revealed that epididymal gene expression is highly segmentary and is a relatively perfect model system. Subsequent studies have shown that spermatogenesis and the expression of immune-related genes *FRA1*, *TEAD3*, *DEFB125*, *SPINT3*, *ACTG2* and *FBLN2* were significantly increased in the epididymal cauda [12]. The expression of key genes *SORD*, *FAM161A* and *MUC1,* related to sperm motility and immune protection, was significantly increased in the epididymal cauda of fertile bulls compared with infertile bulls [10]. Liu et al. [13], showed that the lipid metabolism and antioxidation genes, GPX3, GPX5, *CYCS* and *COX6A1*, were significantly upregulated in the epididymal cauda of Hu sheep with high fertility. In addition, one study found that *Foxa2* expression in the epididymal cauda of mice was significantly upregulated, and this gene had a role in inhibiting androgen activation on lipoprotein 5 [14]. In the rat epididymis, numerous differentially expressed genes (DEGs) were present at different developmental stages and were involved in cell differentiation, immune tolerance, and hormone regulation [15]. In the RNA-seq analysis of the epididymal caput, corpus and cauda of yaks, *MAN2B2*, *MCT7*, *PAG4*, *OAS1* and *TGM3* were highly expressed in the cauda. These genes were associated with post-translational modification and offered an appropriate storage microenvironment for pre-ejaculation [4]. These results suggest that the epididymal cauda has a richer mRNA expression level, may be more widely involved in sperm maturation, and affects the reproductive capacity of males.

Protein is the product of gene transcription and translation and is also a key link in the study of molecular mechanism. A proteomic study revealed that DMRT1, ADGRG2, BSPH1, and other proteins related to lipid metabolism and sperm development were specifically expressed in the human epididymis, and high abundance proteins were significantly enriched in the redox reactions and fatty acid metabolism [16]. Furthermore, albumin, transferrin, and PGDS, which protect sperm from reactive oxygen species (ROS) and maintain sperm fertility, were found to be the main protein components of the caudal lumen fluid of sheep epididymis [17,18]. The amount of protein in the cauda of the mice epididymis is significantly higher than that in the caput of the epididymis. Spink5 is enriched in the cauda of the mice epididymis and has the effect of volatile glycosylation modification, which is involved in the regulation of sperm function [19]. The expression of specific proteins is significantly higher in neonatal epididymis than in the epididymis of young and old people, which indicates that the epididymal protein composition changes dynamically at different development stages [20]. Previous studies have mainly explained epididymal differences over a certain period through a single omics approach. However, a single omics cannot fully explain the molecular mechanism of epididymal physiological changes and sperm maturation influence, and there is still a lack of a genetic map of the cauda epididymis before and after sexual maturation.

Yak (*Bos grunniens*) is a unique cattle species on the Qinghai-Tibet Plateau. It can adapt to this cold and low oxygen plateau environment and is an important functional livestock to meet the production and other living needs of local herdsmen. However, the low fertility of yaks greatly limits the number of yak populations. Therefore, in this experiment, RNA-seq and proteomics were used to screen the key genes of epididymal cauda of 6- and 30-month-old yaks by single and combined analysis methods, so as to provide a theoretical basis for increasing the reproductive potential of yaks and further expanding the number of yak populations.

## 2. Materials and Methods

### 2.1. Epididymal Cauda Sample Collection

All animal-related procedures conformed to the China Council on Animal Care and the Ministry of Agriculture of the People’s Republic of China guidelines, while all yak-handling procedures were approved by The Animal Care and Use Committee of the Lanzhou Institute of Husbandry and Pharmaceutical Sciences Chinese Academy of Agricultural Sciences (Permit No: SYXK-2014-0002). These samples were obtained from the epididymis of 12 well-grown, healthy, pedigree clear male yaks with no reproductive defects in their fathers, in Gannan Tibetan Autonomous Prefecture (103°9′ E, 35°10′ N), located at an altitude of approximately 3500 m (six 6-month-old yaks (Y6m) and six 30-month-old yaks (Y30m)). Before castration, local testis was sterilized with iodophor and alcohol, and after the testicle was removed the wound was sutured. The animals were administered penicillin and streptomycin injection to prevent infection. Using a scalpel, the epididymal tissue was carefully separated from the testicular tissue and, according to the anatomical structure, the epididymis separated into caput, corpora and cauda. Then, the surface tunica was peeled off. The epididymal cauda was washed three times with 0.01 M PBS, transferred to a freezing tube, quickly frozen in liquid nitrogen, brought back to the laboratory and stored at −80 °C for subsequent transcriptome sequence analysis and proteomics analysis. In addition, some epididymal caput, corpora, cauda and testicular tissues were collected for hematoxylin and eosin staining to observe the tissue structure.

### 2.2. Hematoxylin and Eosin Staining 

After fixation, epididymal and testicular tissue samples were dehydrated using 75% ethanol, paraffin-embedded, and cut into 5 mm sections, that were then stained with an improved H&E staining kit (SolarBio, Beijing, China) based on the provided direction. Sections were then sealed using neutral gum and imaged with a Pannoramic 250 digital section scanner (Drnjier, Jinan, China).

### 2.3. RNA Library Preparation and Sequencing 

The TRIzol (Thermo Scientific, Waltham, MA, USA) reagent was used to extract total RNA from the epididymal cauda. RNA purity, quantification, and integrity were assessed using the Agilent 2100 Bioanalyzer (Agilent Technologies, Santa Clara, CA, USA). Then, the libraries were constructed using VAHTS Universal V6 RNA-seq Library Prep Kit (Vazyme, Nanjing, Jiangsu, China) according to the manufacturer’s instructions. The libraries were sequenced on an llumina Novaseq 6000 platform. Raw reads were processed using fastp [21] to de-link, remove low-quality reads, and exclude low-quality bases from 3′–5′ caudas, Q30, and GC content in order to conduct decaudaed evaluation obtain clean reads. The clean reads were mapped to the reference genome (LU_Bosgru_v3.0) using HISAT2 (Johns Hopkins Hospital, Baltimore, USA) [22]. FPKM3 [23] of each gene was calculated, and read counts of each gene were obtained using HTSeq-count4 (http://www-huber.embl.de/HTSeq, accessed on 28 October 2022) [24]. Principal component analysis (PCA) was performed using R (v3.2.0, Shenzhen, China) to evaluate biological duplication of the sample.

### 2.4. Extraction and Pretreatment of Epididymal Cauda Protein

The samples of frozen epididymal cauda were lysed using lysis buffer and through sonication. Then, the samples were centrifuged at 12,000 rpm for 10 min at 4 °C to remove insoluble particles. The supernatants were added at 5 times the volumes of cold acetone and precipitated at −40 °C overnight. The solution was again centrifuged at 12,000 rpm for 10 min at 4 °C to collect precipitates. Once the precipitate dried, it was dissolved in SDS lysis buffer and centrifuged at 12,000 rpm for 10 min to collect supernatants. Using the supernatant, protein concentration was determined through the BCA assay. The proteins were stored at −80 °C for further use.

According to the protein quantification results, 100 μg protein extraction was achieved with 120 μL reducing buffer. The solution was incubated at 60 °C for 1 h, and iodoacetamide solution (final concentration of 50 mM) was added to the solution in the dark for 40 min at room temperature. Then, the solution was centrifuged at 4 °C and 12,000 rpm for 20 min, and the precipitate was washed with TEAB. Later, 3 µL sequencing-grade trypsin (1 µg µL^−1^) was added to the precipitate. Then the solutions were incubated for digestion at 37 °C for 12 h. The solutions obtained were collected and lyophilized.

### 2.5. Reversed-Phase Chromatography and Liquid Chromatography-Mass Spectrometry (LC-MS)

The sample protein labeled by TMTpro was pre-separated by RPC (Agilent 1100, Santa Clara, CA, USA), and Agilent Zorbax Extend-C18 narrow diameter column (2.1 × 150 mm, 5 μm). Using a DAD detector, the UV wavelength was 210 and 280 nm, the mobile phase A was ACN-H_2_O (2:98, *v*/*v*, pH = 10), the phase B was ACN-H_2_O (90:10, *v*/*v*, pH = 10), the elution gradient is shown in Appendix A, and the flow rate was 300 µL min^−1^. Samples were collected for 8–60 min, and eluent was collected in centrifugal tube 1–15 each minute. Samples were recycled in this order until the end of the gradient. The separated peptides were lyophilized for LC-MS.

The pretreated protein was analyzed via LC-MS. The protein sample was loaded onto the precolumn Acclaim PepMap 1000 (100 μm × 2 cm, RP-C18, Thermo Scientific, Waltham, MA, USA), followed by that on the analysis column Acclaim PepMap RSLC separate (75 μm × 50 cm, RP-C18, Thermo Fisher). The mobile phase A was H_2_O-FA (99.9:0.1, *v*/*v*), the mobile phase B was ACN-H_2_O-FA (80:19.9:0.1, *v*/*v*/*v*), the flow rate was 300 nL min^−1^, and the elution gradient used is presented in Appendix A. Full MS scans were acquired in the mass range of 350–1500 *m*/*z* with a mass resolution of 60,000. The AGC target value was set at 3 × 10^6^. The 20 most intense peaks in MS were fragmented through higher-energy collisional dissociation, with a collision energy of 32. MS/MS spectra were obtained at a resolution of 45,000 with an AGC target of 2 × 10^5^ and a max injection time of 80 ms. The Q Exactive HF dynamic exclusion was set for 30.0 s and run in the positive mode.

### 2.6. Screening and Analysis of Differential Genes (DEGs) and Differential Proteins (DEPs)

Differential expression analysis was performed using DESeq2. The *q < 0.01* and |Log_2_FC| > 1 was set as the threshold for significant DEGs. The screening conditions of DEPs were unique peptides ≥ 2, |Log_2_FC| > 1, and *q* ≤ 0.01. Gene Ontology (GO) terms in the database were used for the mapping of DEGs and DEPs between the epididymal cauda tissue of yaks of different months. Bonferroni correction was used to normally adjust the *p* value. KEGG was used for the enrichment analysis of biological pathways of the DEGs and DEPs (*p* < 0.05). To reveal the characteristics of the DEGs, gene set enrichment analysis (GSEA) was performed using GSEA software (Broad Institute, MA, USA). The STRING protein interaction database was used for protein–protein interaction (PPI) network analysis of the DEPs, and the differential PPI network data files were complemented by using Cytoscape software (https://cytoscape.org, accessed on 13 October 2022) for visual editing.

### 2.7. Quantitative Real-Time PCR

Primers were selected according to the RNA-seq results (Appendix A). The LightCycler^®^ 96 Realtime Detection System (Roche, Beijing, China) was used to detect and analyze the selected genes through quantitative Real-Time PCR (qRT-PCR). Glyceraldehyde-3-phosphate dehydrogenase (*GAPDH*) was used as the internal reference gene. The reaction system was 20 μL and comprised 10 μL 2× SYBR Green II PCR Mix (Takara Bio, Beijing, China), 1 μL cDNA (25 ng), 1 μL primers, and 8 µL nuclease-free water. The reaction conditions were as follows: 95 °C for 300 s, followed by 40 cycles at 95 °C for 10 s, 60 °C for 30 s, and 72 °C for 2 min. A melting curve was obtained from 65 °C to 95 °C, which was increasing in increments of 0.5 °C every 5 s. The 2^−∆∆Ct^ method was employed for calculation of relative expression.

### 2.8. 4D-parallel Reaction Monitoring

According to the proteomic results, the selected target protein was validated using 4D-parallel reaction monitoring (4D-PRM) technology. The C18 reversed phase column (75 μm × 25 cm, 1.6 μm, 120 A, C18, IonOpticks) was used. The mobile phases A and B were ACN-H_2_O-FA (0:100:0.1, *v*/*v*/*v*) and ACN-H_2_O-FA (100:0:0.1, *v*/*v*/*v*), respectively. The flow rate was 300 nL min^−1^, and elution was performed using a multistep gradient (0–45 min, 2–22% B phase; 45–50 min, 22–37% B phase; 50–55 min, 37–80% B phase; 55–60 min, 80% B phase. TimsTOF Pro2 used the parallel cumulative continuous fragmentation PRM mode for data acquisition in the positive ion mode. The capillary voltage was established at 1400 V, the primary and secondary sweep range of the mass spectrum was 100–1700 m/z, and the ion mobility window range (1/K0) was 0.6–1.6 Vs/cm^2^. Ion accumulation and release time was set at 100 ms to achieve close to 100% ion utilization. The time window was set as 10 min, with the ion migration resolution being 50.

### 2.9. Statistical Analysis

The data were integrated by Excel 2016 and analyzed by SPSS 22.0 (IBM, NY, USA), and the results showed as Mean ± SEM.

## 3. Results

### 3.1. Histological Observation of Epididymis and Testis

The epididymal and testicular tissues were histologically observed under an optical microscope (10×, Figure 1A–D), and the parts magnified showing obvious characteristics (40×, lower right corner). The results showed that there were obvious differences between the epididymal caput, corpus, cauda and the testis. The testicular lumen is the smallest and the number of cell layers on the wall is the largest. The lumen diameter in the caput of the epididymis is large and the number of lumens is dense. The lumen diameter of the epididymis corpus is small, while the lumen diameter of the cauda of the epididymis is large and there are a lot of sperm.

### 3.2. Summary of RNA-Seq Data

To obtain the transcriptional map data of the epididymal cauda of yaks of different ages, we sequenced these tissues through RNA-seq and filtered them using Trimatatic v0.36 software (http://www.usadellab.org/cms/index.php?page=trimmomatic, accessed on 28 October 2022). In total, 77.13 G effective bases were obtained, accounting for 93.13% of the total number of bases, with an average of 6.43 G per sample (Appendix A). The gene expression level in each sample was estimated on the basis of the FPKM value (Appendix A, Figure 2A,B). The genes at different expression levels in the same group of samples were similar, and the total gene level was consistent. The PCA results revealed no aggregation between different groups, but the aggregation between the same group was tight (Figure 2C). Correlation and cluster analyses revealed significant differences between different groups, but small differences within groups. This indicated that the sample repeatability within the same group was high (Figure 2D).

### 3.3. Identification of DEGs

According to our screening conditions (*q* ≤ 0.01 and |Log_2_FC| > 1), we screened 2274 DEGs; of these, 961 were upregulated and 1313 were downregulated genes (Appendix A, Figure 2E). After clustering, the DEGs were significantly divided into two groups, indicating a significant difference in gene expression between the two sample groups (Figure 2F). Our analysis of 30 genes with the most significant differences demonstrated that seven genes were upregulated and 23 genes were downregulated. The upregulated genes were mainly involved in cell differentiation, proliferation and the cell cycle, and included glypican 3 (*GPC3*), microfibril associated protein 2 (*MFAP2*), prokineticin receptor 1 (*PROKR1*), and HtrA serine peptidase 3 (*HTRA3*). The downregulated genes included sodium channel epithelial 1 subunit alpha (*SCNN1A*), glyceraldehyde-3-phosphate dehydrogenase, spermatogenic (*GAPDHS*), creatine kinase mitochondrial 1 (*CKMT1*), nuclear RNA export factor 3 (*NXF3*), and stomatin-like 1 (*STOML1*). These genes are associated with spermatogenesis, development and microenvironment construction. This also included related genes involved in immune response, including surfactant associated 2 (*SFTA2*), basal cell adhesion molecule (*BCAM*), solute carrier family 16 member 6 (*SLC16A6*), uroplakin 2 (*UPK2*), formin-like 1 (*FMNL1*), and glucuronidase beta (*GUSB*). In addition, genes encoding methionine adenosyl-transferase 1A (*MAT1A*) and family with sequence similarity 221 member A (*FAM221A*), which were possibly related to methylation modification, were also present.

### 3.4. GO, KEGG Pathway and GSEA Enrichment Evaluations for DEGs

Results of GO, KEGG, and GSEA analyses of DEGs were evaluated to describe the function of differential genes in the yak epididymal cauda at different developmental stages. Based on the GO analysis, DEGs were divided into biological processes (BP), cellular components (CC), and molecular functions (MF), displaying significant enrichment in 645 GO items after screening. The most significant differences in BP analysis were noted in cell adhesion (upregulated) and spermatogenesis (downregulated). The most significant CC were the extracellular matrix (upregulated) and acrosomal vesicle (downregulated). The MF included the extracellular matrix structural constituent (upregulated) and acyl-CoA hydrolase activity (downregulated) (Appendix A, Figure 3A,B). Moreover, spermatid development, flagellated sperm motility, and sperm capacitation were significantly downregulated in the GO analysis.

The upregulated DEGs are significantly enriched in ECM-receiver interaction, Wnt signaling pathway, PI3K Akt signaling pathway, and Hedgehog signaling pathway in KEGG. The downregulated DEGs are significantly enriched in lysosome, ABC transporters, sphingolipid metropolis, and steroid household biosynthesis (Appendix A, Figure 3C,D). Furthermore, we selected axon guidance (upregulation) and lysosome (downregulation) signaling pathways in KEGG for GSEA, and the results were consistent with those of KEGG enrichment (Figure 3E,F).

### 3.5. Overview of Proteome Sequencing Data and Screening and Identification of DEPs

Using the TMTpro labeling quantitative technique, we obtained the results of DEPs in the epididymal cauda of yaks of different ages. In total, 7879 pieces of original protein data were obtained and 5828 peptides, after screening according to credible proteins (unique peptides ≥ 2) (Appendix A). The PCA results revealed no aggregation between samples of different groups, indicating that the grouping was good (Figure 4A).

With further screening (|Log_2_FC| > 1 and *q* ≤ 0.01), 222 DEPs were obtained; of these, 87 were upregulated and 135 were downregulated (Appendix A, Figure 4B). After clustering, DEPs were significantly divided into two groups, which indicated a significant difference between the two sample groups (Figure 4C). We sorted DEPs according to the q value from small to large and analyzed the first 30 DEPs. Of these, 15 were upregulated proteins and 15 downregulated proteins. The upregulated proteins included integrin alpha-4, tubulin beta chain, peroxidasin-like protein, cellular retinoic acid-binding protein 1, neuronal cell adhesion molecule, glypican-6, and methanethiol oxidase. These proteins are related to cell proliferation, differentiation and adhesion. In addition, the upregulated protein dopamine beta-hydroxylase can participate in norepinephrine synthesis regulation. The downregulated proteins were related to spermatogenesis, maturation and sperm microenvironment construction and included chloride anion exchanger, 45-kDa calcium-binding protein and creatine kinase. The immune-related proteins tripeptidyl-peptidase 1, galectin, beta-defensin and beta-mannosidase were also significantly upregulated. Moreover, the protein S-adenosylmethionine synthase, which is related to methylation modification, was also upregulated.

### 3.6. GO and KEGG Enrichment Analyses of DEPs in Epididymal Cauda

Biological analysis was performed on the selected epididymal cauda DEPs to reveal the functional components and pathways involved. Enrichment of upregulated DEPs revealed that the collagen biosynthetic process (BP), neurofilament (CC), and oxygen carrier activity (MF) were significantly enriched in GO. According to KEGG, the ECM–receptor interaction was significantly enriched (Appendix A, Figure 5A,B). Enrichment of downregulated DEPs revealed that the acyl-CoA metabolic process (BP), lysosome (CC), and alpha-mannosidase activity (MF) were significantly enriched in GO. According to KEGG, peroxisome, lysosome, biosynthesis of unsaturated fat acids, and steroid biosynthesis pathways were significantly enriched (Appendix A, Figure 5C,D).

### 3.7. Protein–Protein Interaction (PPI) among the DEPs

DEPs were searched with different manifestations through STRING (www.string-db.org, accessed on 16 November 2022) and a visual network diagram drawn for interacting proteins. The minimum required interaction score was set as 0.400. The PPI enrichment *p* value was 1.17 × 10^−11^ and 1.0 × 10^−16^, and Cytoscape 3.9.0 was used to optimize the protein network, which could be divided into four clusters. The results revealed that the upregulated DEPs were clustered into Cluster 1 and Cluster 2 (Figure 6). These DEPs were involved in protein metabolism, cell proliferation, differentiation, adhesion, and cell cycle regulation. The downregulated DEPs were clustered in Cluster 3 and Cluster 4. These DEPs were mainly related to steroid hormone synthesis and sperm maturation modification (Cluster 3), as well as immune defense and inflammation protection (Cluster 4).

### 3.8. Combined Analysis of Transcriptome and Proteome

According to the previously screened DEGs and DEPs, the correspondences of correlation and differential data between RNA-seq and proteomics were calculated according to the one-to-one correspondence between samples (Appendix A, Figure 7). The results revealed that a good correlation existed between DEGs and DEPs (Figure 7A). The expression of 3719 screened genes were shown in the nine-quadrant diagram; of these, 60 were co-upregulated, 49 were co-downregulated, and eight were inversely regulated (Figure 7B).

### 3.9. Analysis of GO and KEGG Genes with the Same Expression Trend

GO and KEGG enrichment analysis was performed on all co-expressed genes. The results showed that, in GO enrichment, all up-regulated co-expressed genes were mainly enriched in collagen fibril organization, extracellular matrix, and cell adhesion (Appendix A, Figure 8A). The KEGG enrichment was mainly for protein digestion and absorption, via the ECM-receptor interaction pathway (Figure 8B). GO enrichment was mainly in intermediate filament, structural molecule activity and lysosome (Appendix A, Figure 8C). KEGG enrichment showed lysosome and ferroptosis (Figure 8D).

### 3.10. Verification of Differential Genes and Differential Proteins through qRT-PCR and 4D-RPM

The RNA-seq and proteomic results were verified through qRT-PCR and 4D-PRM, and 14 differential genes and 20 differential proteins were selected. The results revealed that the qRT-PCR results were consistent with RNA-seq data (Figure 9A), and the expression trends of 4D-PRM and proteome genes were the same (Figure 9B). The results of RNA-seq and proteomics were reliable.

## 4. Discussion

The epididymal cauda is an important place for sperm maturation and storage [25]. However, there is still a gap in the study of the epididymal cauda before and after sexual maturity in yaks [2,4,26]. We here analyzed the top 30 significant DEGs obtained from RNA-seq and found that upregulated DEGs were mainly related to cell proliferation and differentiation, while the downregulated DEGs were mainly related to sperm maturation and modification. The differentially most significantly upregulated gene *GPC3* activates the typical WNT/β-catenin pathway through its core protein and heparan sulfate to regulate cell growth and differentiation. This gene is also involved in regulating body growth and development through the regulation of insulin-like growth factor 2 (IGF2) and the Hedgehog signaling pathway [27,28]. In addition, when fetal and adult kidneys were examined, *GPC3* was significantly expressed in the fetus, similar to our results [29]. *GPC3* was significantly upregulated in 6-month-old yaks, but its potential function in the epididymal cauda requires further exploration. The significantly downregulated gene *GAPDHS* can continuously catalyze the sperm glycolysis pathway and enable sperm to achieve motility during maturation through phosphorylation of sperm protein phosphatase 1 [30]. In *GAPDHS* knockout mice, the fine structure of the fibrous sheath of sperm was damaged, sperm motility had disappeared, and sperm had become infertile [31]. Immunohistochemistry of sperm exhibiting flagellum dysplasia revealed that *GAPDHS* was distributed abnormally and irregularly on the flagellum, indicating that *GAPDHS* has a crucial role in promoting sperm maturation in the epididymis [32]. However, no significant difference was observed between *GPC3* and *GAPDHS* in proteomic detection. Limited overlap between protein and mRNA data was also observed in other studies on humans, cattle, and mice, which may be related to differences in experimental techniques, shortening of 3′UTR, and differences in spatiotemporal expression [16,33].

In the conjoint analysis, a total of 109 identical trend genes was observed (60 were co-upregulated, 49 were co-downregulated), mainly concentrated in relation to cell development, sperm maturation, and immune defense and included TGFBI, stmn1-a, Septin3, COL1A1, COL1A2, SULT2B1, MANBA, KRT19, and NPC2. Factor beta-induced (TGFBI) can mediate the binding of the ECM protein and plays a role in cell adhesion and proliferation in various cell environments. TGFBI can also promote the glycolysis pathway and increase the cell migration potential and trend ability through the PI3K signal pathway [34]. In a study, TGFBI expression was significantly increased in the uterus and placenta of a mouse pregnant for 18 days, and TGFBI was involved in regulating the invasiveness of trophoblast cells [35]. Furthermore, TGFBI defects also destroy the alveolar infrastructure and reduce muscle development and cell attachment [36]. TGFBI also has an arginine-glycine-aspartic sequence. This sequence can bind with the ECM receptor integrin, activate the AKT/MAPK signal pathway, and affect the adhesion and migration of vascular smooth muscle cells, thereby regulating and controlling abnormal angiogenesis, TGFBI also promotes endothelial cell proliferation and migration through interaction with VEGFR2 [37,38]. However, studies have shown that TGFB1 remains unchanged during rat epididymal development, and the TGFB pathway may be related to sperm immunity [15]. This is different from our results, suggesting that TGFB1 may play different roles in the development of epididymis in different species, which requires further exploration of the specific regulatory mechanism of TGFB1.

The collagen family is a major component of the ECM and participates in the development and composition of various tissues and organs. In the present study, collagen family members COL1A1, COL1A2, COL3A1, and COL12A1 were significantly enriched in the ECM–receptor interaction. COL1A1 and COL1A2 expression in the vas deferens was significantly higher in 6-day-old mice than in 60-day-old mice and had a potential role in cell adhesion [39]. However, COL1A1 and COL3A1 exhibited different regulation modes during bone development, which is possibly because COL3A1 is the main structural component of hollow organs [40]. COL3A1 can improve cell stretch resistance and integrity and exhibits different expression patterns in mice of different ages [41]. COL12A1 is significantly expressed in loosely attached embryos. In addition to regulating cell adhesion and migration, COL12A1 is involved in maintaining the structural integrity of the cumulus–oocyte complex [42,43]. This significantly expressed collagen family member may be widely involved in the development of epididymal cauda tissue and promote gonadal maturation.

Downregulated DEPs were significantly enriched in the steroid hormone biosynthesis pathway, whereas in the conjoint analysis they were specifically described as enriched in the estrogen signaling pathway. In rats, the undifferentiated epididymis was significantly associated with thyroid hormones, kidney development-related genes, and other regulatory factors, while estrogen- and androgen-related pathways and androgen-dependent transcription factors were abundant during epididymal dilation and differentiation, similar to our results [43,44]. Sulfotransferase family 2B member 1 (SULT2B1) can sulfonate various steroids, which is considered an effective means of hormone homeostasis in the testis [45]. In the comparative analysis of porcine SULT2B1, it was found to be significantly expressed in the epididymal cauda, and its expression increased along the direction of sperm flow, namely testis < epididymis caput < corpus < cauda; this specific expression contributes to the stability of the sperm cell membrane and inhibits acrosin to maintain sperm motility [46]. As a supplement, this experiment revealed that SULT2B1 expression in the epididymal cauda varied in the different stages. Keratin 19 (KRT19) was significantly expressed in high-fertility bull sperm; this high KRT19 expression is speculated to be associated with acrosomal integrity and functional membrane integrity and may also be involved in placental development after fertilization [47]. In particular, KRT19 only has androgen response elements (ARE). Combined with mice experiments, we speculated that KRT19 is activated with an increase in androgen expression in the middle and late stages of gonadal development. Moreover, it is involved in sperm development, affects the estrogen signal pathway through the non-estrogen receptor pathway, and participates in sperm development and maturation [48,49].

In conjoint analyses, the lysosomes pathway was significantly enriched. NPC internal cholesterol transporter 2 (NPC2) is a crucial participant in the lysosomal signaling pathway [50]. NPC2 expression was significantly reduced in juvenile and senescent canine sperm and was associated with decreased sperm quality [51]. Studies in different epididymal regions of adult bulls have revealed that NPC2 expression significantly increased in the epididymal cauda and is one of the most abundantly expressed proteins. Moreover, it is involved in sperm cholesterol outflow and sperm maturation and could be used as a membrane recombination factor in the independent capacitation process [52,53]. After vasectomy, NPC2 content in the epididymis decreased, resulting in abnormal sperm biochemical modification and changes in raft membrane domain content and thus affecting sperm capacitation. In addition to sperm modification, lysosomes also play other roles, including sperm autophagy [54]. The inhibition of human sperm autophagy under overexposure to ROS leads to deterioration of sperm quality and metabolic parameters and increase in cell death markers [55]. Interference with autophagy leads to abnormality in sperm head and flagella function [56]. Moreover, lysosomes are associated with sperm modification and storage and the construction of the epididymal lumen microenvironment [57]. It has been reported that acrosomes originate from autolysosomes and affect acrosomal reactions in sperm [58,59]. Along with the present study results, we speculated that lysosomes were significantly enriched in the epididymal cauda of sexually mature yaks. In addition to sperm modification and protection from oxidative stress, degraded or abnormal sperm could also be dissolved and absorbed through autophagy. However, the specific regulatory mechanism remains unclear and needs to be investigated.

## 5. Conclusions

In this experiment, RNA-seq and proteomics were used to analyze the expression of key genes in the epididymal cauda of yaks before and after sexual maturity. All up-regulated co-expressed genes were mainly related to cell growth, differentiation and adhesion. and thus, they promoted before sexual maturity the development of gonads and reproductive organs. All down-regulated co-expression genes were mainly related to steroid synthesis, sperm modification and maturation. Abnormality of these genes may lead to abnormal epididymal cauda function after sexual maturity and affect sperm maturation. In conclusion, this study provides valuable information related to epididymis development, the mechanism of sperm maturation, and the improvement of the reproductive efficiency of yaks.

## Figures and Tables

**Figure 1 animals-13-01355-f001:**
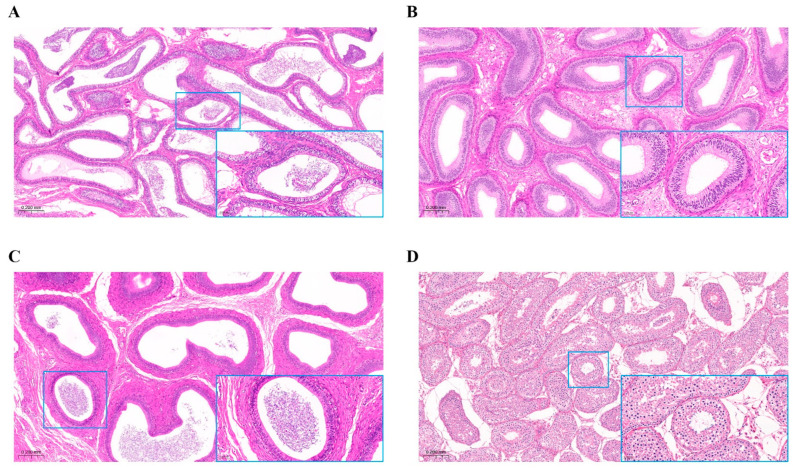
Histological observation of epididymis and testis. (**A**) Epididymis caput. (**B**) Epididymis corpus. (**C**) Epididymis cauda. (**D**) Testicular tissue. The blue box shows a partial enlarged (40×) view of the organization.

**Figure 2 animals-13-01355-f002:**
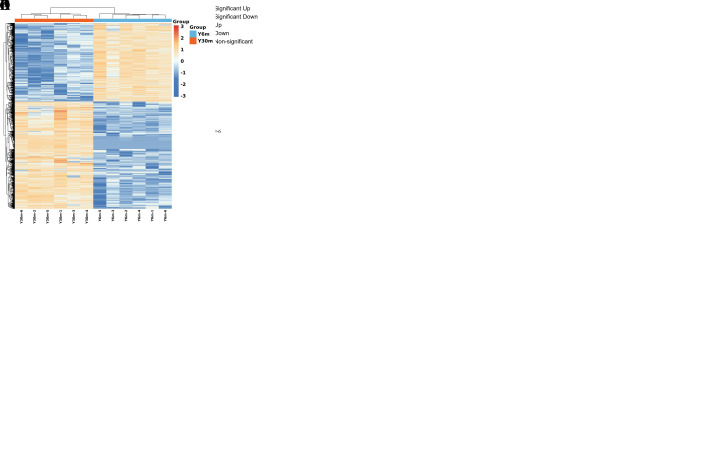
RNA-seq data summary. (**A**) The expression distribution of FPKM in each sample. (**B**) FPKM violin diagram. (**C**) PCA diagram. (**D**) Correlation heatmap and cluster analysis among samples. (**E**) Gene volcano map of the difference between 6m and 30m groups. (**F**) Cluster analysis of different gene groups. Y6m: 6-month-old group. Y30m: 30-month-old group.

**Figure 3 animals-13-01355-f003:**
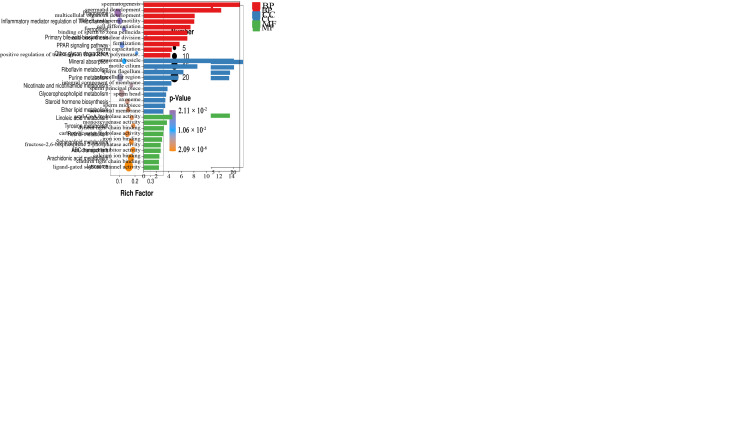
GO, KEGG, and GSEA analyses for DEGs. GO enrichment analysis of DEGs showed that (**A**) was upregulated and (**B**) was downregulated. KEGG enrichment analysis of DEGs showed that (**C**) was upregulated and (**D**) was downregulated. GSEA enrichment results revealed upregulation of (**E**) and downregulation of (**F**). GO enrichment analysis ranked 10 entries from large to small according to the respective −log10 *p* value. KEGG analysis exhibited some significant enrichment results. GSEA analysis selected the top KEGG for enrichment analysis verification.

**Figure 4 animals-13-01355-f004:**
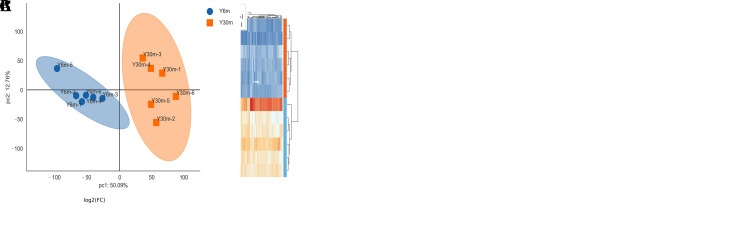
Overview of proteome sequencing data. (**A**) PCA diagram of two sample groups. (**B**) Differential protein volcano map. (**C**) Cluster analysis of differential proteins.

**Figure 5 animals-13-01355-f005:**
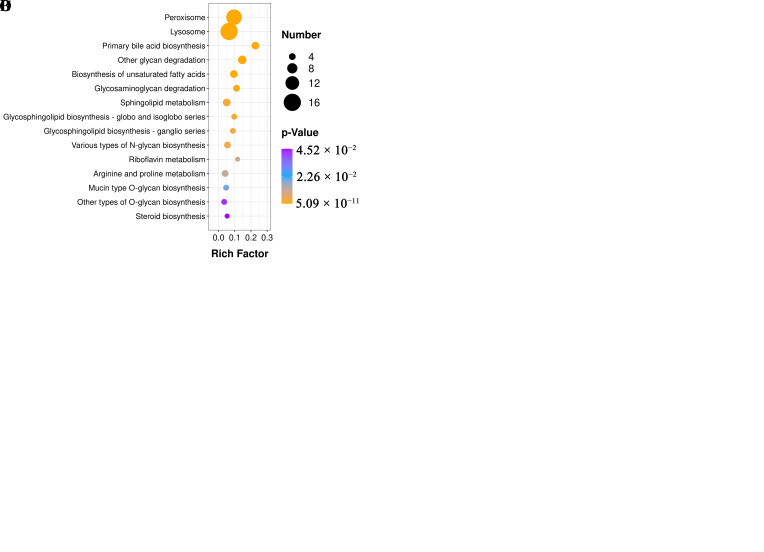
GO and KEGG enrichment analyses. (**A**) GO enrichment of upregulated DEPs. (**B**) KEGG enrichment analysis of upregulated DEPs. (**C**,**D**) GO and KEGG enrichment analyses of downregulated DEPs.

**Figure 6 animals-13-01355-f006:**
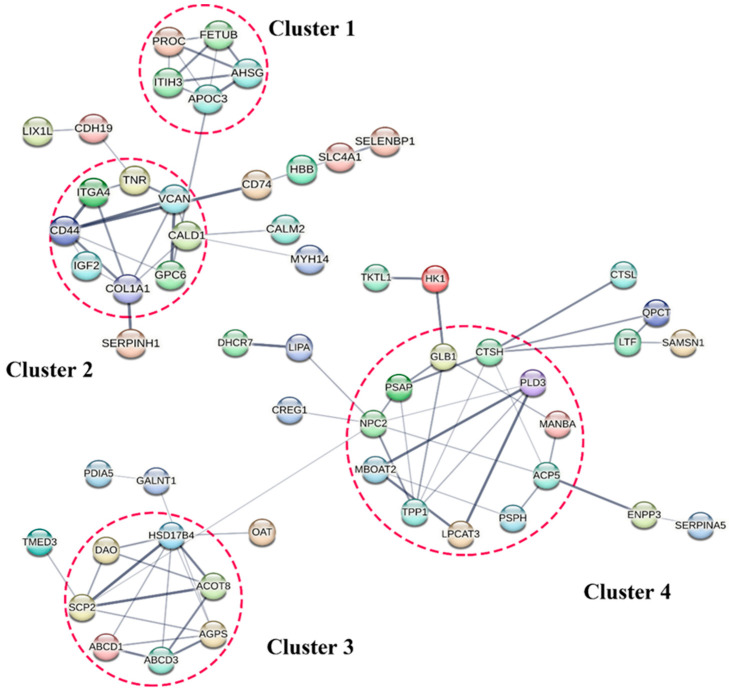
Network diagram of differential protein interaction in epididymal cauda. The protein–protein interaction network diagram was drawn using STRING. Cluster 1 and Cluster 2 represented upregulated DEP aggregation. Cluster 3 and Cluster 4 were formed when DEPs were downregulated.

**Figure 7 animals-13-01355-f007:**
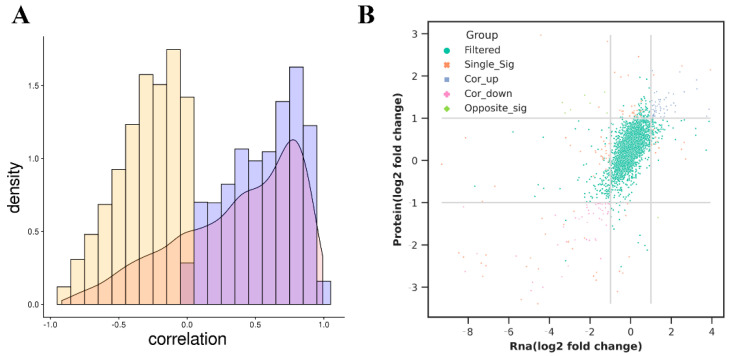
Integration analysis. The differential genes and differential proteins were analyzed together. (**A**) presents the correlation between DEGs and DEPs. (**B**) presents the nine-quadrant diagram.

**Figure 8 animals-13-01355-f008:**
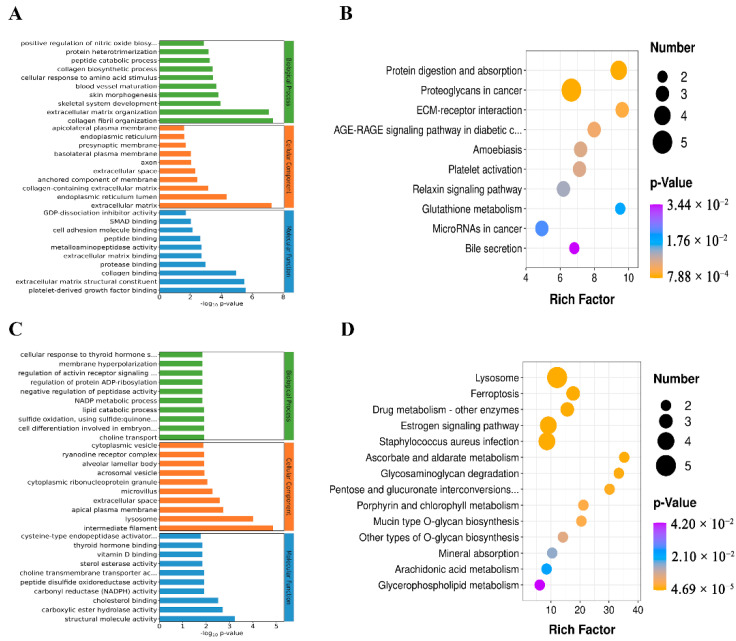
Analysis of GO and KEGG genes with the same expression trend. (**A**,**B**) present the GO and KEGG enrichment of co-up-regulated genes. (**C**,**D**) present GO and KEGG enrichment analyses with downregulated expression of proteins and mRNA.

**Figure 9 animals-13-01355-f009:**
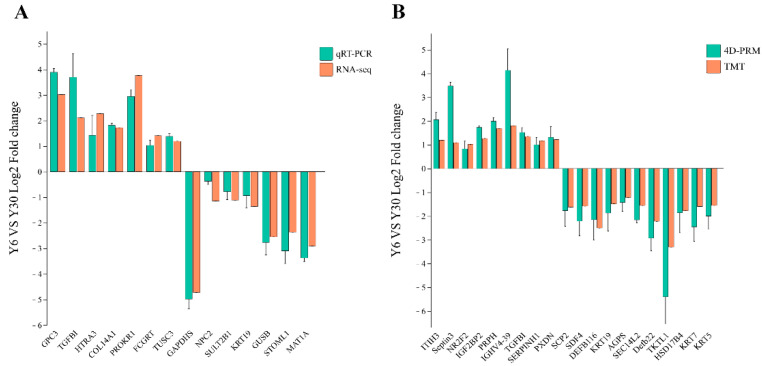
Differential genes and differential proteins were verified through qRT-PCR and 4D-PRM. (**A**) Differential genes were verified through qRT-PCR, and the results were presented as Log_2_FC. (**B**) The selected differential proteins were validated through 4D-PRM.

## Data Availability

All RNA-seq and proteomic data used in this study are available in the SRA database (ID: PRJNA931450, https://www.ncbi.nlm.nih.gov/sra/PRJNA931450, accessed on 18 February 2023) and Proteome Xchange database (ID: PXD039871, http://proteomecentral.proteomexchange.org/cgi/GetDataset?ID=PXD039871, accessed on 3 February 2023). Other raw datasets may also be requested from the corresponding author provided that all ethical requirements have been met.

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
