# Peer review of "Comparative Analysis of Epididymis Cauda of Yak before and after Sexual Maturity"

_animals, 2023, doi:10.3390/ani13081355_

Round 1

Reviewer 1 Report

The manuscript is quite relevant and interesting for specialists. But I would like to make a number of remarks.

Why is the selection of sperm from the epididymis not done? It would be possible to assess the quality of sperm, such as quantity, motility and morphology.

Without this evaluation, the manuscript loses

Reviewer 2 Report

This is a very interesting paper with significant implications for understanding and improvement of yak reproduction. Epididymis development is the basis of male reproduction. And it is also a crucial site for sperm maturation.  This study investigates the differential gene expression during different stages of cauda epididymis before and after sexual maturity in yak and tries to understand the epididymal development of yak, as well as how sperm maturation is regulated.  The important part of this research is that the authors utilized multi-omic approaches. The authors analyzed not only the transcriptomic but also the proteomic. The data were adequately analyzed, and the manuscript was well written. Some minor points require improvement.

1, there are some grammar mistakes, such as line 47, “these transcript…” should be these transcripts or this transcript…, There are more. Please check the manuscript carefully.

2, most of the methods are clearly described. But there is some confusion. 

Line 149: iodoacetamide solution (100 mM) was added to …, is it the stock or the final concentration? It looks is the stock solution concentration; what is the final concentration?

Line 153: “The cells were digested…” What cells? This paragraph is talking protein extraction and cells were lysed.

Reviewer 3 Report

Dear Authors,

The study represents numerous novel information about proteomic and genomic aspects of the reproduction of Yak bulls. Especially pre- and post-puberty comparisons are gained the key significance to the study.

However, there is some missing information about the methods for selecting bulls:

Firstly, it is important to explain well the bull selection criteria. How could you determine the bulls for enrolling in the study? It is understood that the only feature was to be healthy. But if it is a reproduction study, more criteria are needed for selection like andrological specialties, breeding values, testicle diameters, and semen characteristics of the mature ones, etc. On the other hand, if it is only a study that explores and identifies the proteomic situation of Yak bulls, the material number could be insufficient. The statistical method to determine the number of animals must be indicated.

The second is about the aims and the conclusions of the study. You mentioned that the results of the study would provide reproductive efficiency to produce Yaks and increase the reproductive potential of them. You must explain how the study can realize these expectations. Because the study seems only a theoretical proteomic and genomic exploration study and it does not reveal any reproductive features except the organs which are interested. It is true that the study results have the potential to be integrated into clinical applications, but you should mention how it can be realized.

Reviewer 4 Report

1. Title of the Manuscript

"Comparative analysis of epididymis cauda of yak before and 2 after sexual maturity"  

The title suitably represents the content in the manuscript.

2. Abstract

Abstract is very well presented containing hypothesis, experimental design, results and conclusion.

3. Introduction

Introduction has aptly presented the review of literature, the gaps in knowledge, hypothesis and objectives.

Few very important studies have been left out related to this work which needs to inserted. 

Kumar, A., Yadav, B., Swain, D.K., Anand, M., Madan, A.K., Yadav, R.K.S., Kushawaha, B. and Yadav, S., 2020. Dynamics of HSPA1A and redox status in the spermatozoa and fluid from different segments of goat epididymis. Cell Stress and Chaperones25(3), pp.509-517.

4. Materials and Methods

Very well explained.

5. Results

Results have been written using good tables, figures etc.

6. I could observe that there are few studies which are used for discussion purpose but no way related with either male or female reproduction. I feel that related studies with relevance should be discussed.

7. Conclusion; Perfect   
